

# Volumetric comparison of mandibular condyles and mandibles in the different skeletal classes in the Saudi population

Hussain Y. A. Marghalani

Department of Orthodontics, Faculty of Dentistry, King Abdulaziz University, Jeddah, Saudi Arabia

## ABSTRACT

**Background:** Assessing the relationship between the condyle and mandible volume and the various skeletal classes is essential in orthodontic diagnosis. The current study evaluated this relationship using volumetric cone-beam computed tomography (CBCT), cephalometric methods, and the correlations between them.

**Materials and Methods:** The study examined 37 full-head CBCTs (74 condyles) from adults in the Saudi population. The condyle and mandible were separated from within the CBCT images. The volume of each segment was compared to measurements from multiple cephalometric analyses.

**Results:** The combined total condylar volume has a moderate correlation with the maxillomandibular differential in each of the genders and in the total sample. Mandibular volume has a significant correlation with the Wits appraisal (sagittal classification) in males. It was also significantly correlated with the vertical classification using gonial angles in females and in the total sample.

**Conclusion:** The relationship between mandible and condyle volume and cephalometric measurements is both dimensional and within the maxillomandibular complex rather than positional or related to the cranial base. Also, the correlation between the condylar and mandibular volumes and the sagittal and vertical dimensions in the orthodontic skeletal classes provides better insight into the mandibular complex.

# INTRODUCTION

The mandible has a particular role in both craniofacial evaluation and orthodontic treatment planning. Cephalometric analyses of the mandible are used to evaluate the individual's growth pattern and to aid in the classification of skeletal form, in both sagittal and vertical directions (*Jacobson, 2006*). Classifying patients using these two directional methods helps in understanding the patient's growth and also helps formulate orthodontic treatment objectives (*Kim et al., 2005*). In addition to the analysis of the mandible in determining skeletal classification, condyle analysis may be related to these different skeletal patterns as well.

Corresponding author
Hussain Y. A. Marghalani,
hymarghalani@kau.edu.sa

The condyle is not only a part of the mandible; the condylar cartilage is assumed to affect mandibular growth (*Meikle, 1973*). As pressure is exerted on the condyle into the fossa, such as when the mandible is retracted, the growth of the condyle is restricted (*Von den Hoff & Delatte, 2008*). If the pressure is relieved from the condyle, such as when the mandible is protracted, the condyle is unrestricted (*Enlow & Hans, 1996*). In order to study the condyle and mandible in relation to skeletal class characteristics, the cephalometric technique has been used as the main method of investigation. For example, cephalometry was used to inspect condylar position, angle, and height in Class II Division 2 malocclusion (*Katsavrias, 2006*) and to investigate condylar head inclination in the vertical dimension (*Gowda et al., 2013*).

The advent of CBCT enabled better evaluation of the mandibular structures (*Garcia-Sanz et al., 2017*; *Saccucci et al., 2012*). Using this technology allowed for 3D volumetric measurements of the condyle (*Garcia-Sanz et al., 2017*) including use in cases of pathological diseases such as unilateral condylar hyperplasia (*Nolte et al., 2016*), and use in treatment planning in orthodontics (*Agrawal et al., 2013*) and orthognathic surgeries (*Cevidanes et al., 2005*). Various studies utilized CBCT to examine the volume of the condyle and mandible, which was found to provide accurate assessment of these structures (*Bayram et al., 2012*; *Kim et al., 2021*). Condylar and mandibular volumes were assessed in cases with mandibular asymmetry (*Hikosaka et al., 2023*). This modality was utilized to investigate the changes in mandibular bone volume in periodontally compromised patients who had full-arch implants (*Kubica et al., 2022*) and to assess the volume and quality of the bone taken from the mandibular ramus in autologous graft procedures (*Kadkhodazadeh et al., 2022*).

Recently, CBCT technology was used to study the different condylar and mandibular dimensions in the various skeletal classes in untreated cases (*Fan et al., 2021*), surgically-treated cases (*Zupnik et al., 2019*), or non-surgically-treated cases (*Al-Saleh et al., 2015*). In various research, different aspects of the relationship between the condyles and cephalometric measurements were explored in both sagittal and vertical dimensions. ANB value, measured in the sagittal dimension (*Ceratti et al., 2022*; *Loiola et al., 2023*; *Saccucci et al., 2012*), mandibular plane angle (*Ceratti et al., 2022*; *Saccucci et al., 2012*), and gonial angle (*Ceratti et al., 2022*), both measured in the vertical dimension, were studied in relation to condylar volume.

In this study, the null hypothesis states that there is no relationship between both the condylar and mandibular volume and the skeletal classes in the sagittal or vertical dimensions. The alternative hypothesis is that condylar and mandibular volumes are related to the skeletal classes that are represented by sagittal and vertical cephalometric measurements. The objectives of this study are to determine whether condylar volume and mandibular volume are related to the skeletal classes, and to investigate factors that might contribute to this relationship between condylar and mandibular volume and the different skeletal classes. These objectives were examined in both sagittal and vertical dimensions.

## MATERIALS AND METHODS

The study was retrospective and cross-sectional. Samples were collected from the Oral Radiology department at King Abdulaziz University Faculty of Dentistry and are representative samples of the Saudi population. Obtaining patient consent was not applicable due to the retrospective nature of the study. Ethical approval was given by the Research Ethical Committee (REC) at the Faculty of Dentistry in King Abdulaziz University (No. 075-03-19). To select the patient population, stringent inclusion and exclusion criteria were utilized. The inclusion criteria incorporated the following: adult patients, Saudi nationality, CBCTs showing the complete mandible, nasion, and sella turcica, and normal bone architecture (no severe bone resorption or recent extraction sockets). The exclusion criteria included the following: craniofacial deformity such as cleft palate patients, maxillomandibular malformations such as condylar hyperplasia, dental anomalies such as mesiodens, edentulous patients, jaw fractures, presence of orthodontic treatment or orthognathic surgery, bite-blocks, rapid palatal expanders, or implants. CBCTs with the following features were excluded: small field of view or not showing the full head, blurry images, evidence of severe malposition during acquisition of the CBCT, condyles or mandible images that were cut, supernumerary teeth, and the presence of surgical plates.

For acquiring CBCT images, iCAT (Imaging Sciences International) was used at 120 Kvp and 5 mA. Figure 1 shows the segmentations of the mandible and the condyles which were taken from the CBCT using Invivo Plus software (Version 6.5; Osteoid Inc, Santa Clara, CA, USA). First, the mandible was segmented from the CBCT and the crowns of the teeth were removed. Each side of the mandible was made visible one at a time to be able to accurately segment the right and left condyles. The condyle was segmented using landmarks from the guidelines of the AOCMF classification system (*Neff et al., 2014*). A line that is perpendicular on the posterior border of the mandible and tangent to the deepest area of the sigmoid notch was used to segment the condyles (Figs. 1A–1C).

The CBCTs were reconstructed into lateral cephalograms using Invivo Plus software. They were then imported into Dolphin Imaging software (Version 11.95; Dolphin Imaging and Management Solutions, Chatsworth, CA, USA) for cephalometric tracing and analysis (Fig. 1D). The cephalometric measurements (Fig. 2) taken were in the sagittal dimension: SNB, ANB, Wits appraisal, nasion perpendicular to Pog, effective mandibular length and maxillomandibular differential. For the vertical dimension, the following measurements were taken: Y-axis, mandibular plane angles (FH-GoMe, SN-GoGn, FH-GoGn, MP-SN), lower anterior facial height, and gonial angles with different landmarks as demonstrated in Table 1.

To test intra-examiner reliability, a single trained investigator (H.M.) took measurements at two timepoints at least 2 weeks apart. Cronbach's Alpha was computed for all the variables. Almost all the values were greater than 0.9 (range 0.76–0.99) indicating good intra-examiner reliability. Sample size calculation was performed with a minimum power of 0.80, Spearman correlation parameter of 0.5, and zero null value by

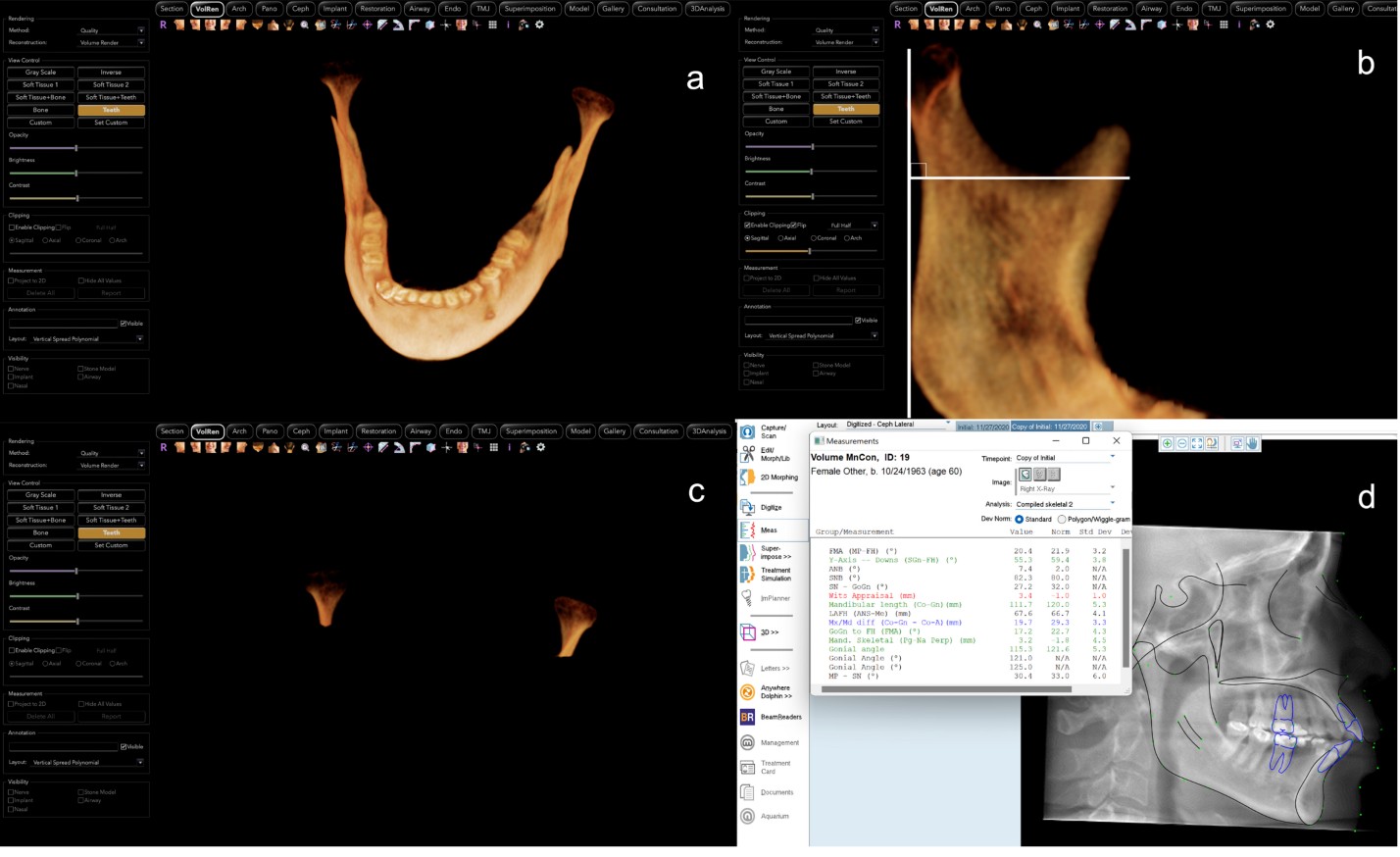

**Figure 1 CBCT segmentations and cephalometric measurements.** Example of segmentations and measurements used in the study: (A) 3D segmentation of the mandible using Invivo software, (B) the method used to segment the condyle, (C) the condyles after segmentation from CBCT image and (D) measurements taken using Dolphin software.

Bonnet and Wright's proposed method using non-directional analysis with a significance level of 0.05. This resulted in a sample of 33 cases.

Data analysis was performed using the Statistical Package for Social Sciences (SPSS Version 29; IBM Corp, Armonk, NY, USA). The distribution of the data was not normal as verified using the Shapiro-Wilk test. Hence, non-parametric tests were used for inferential data analysis. Mann-Whitney U test was used to study the difference between the means in relation to the gender of the study participants. Correlation between the variables was calculated using the Spearman Rank correlation. All *p*-values less than 0.05 were considered statistically significant. The classification of the magnitude for correlation values by Evans was used (*Evans, 1996*).

## RESULTS

A pool of 829 cases was screened using the above mentioned inclusion and exclusion criteria. After applying the strict criteria, the following cases were excluded: CBCTs not showing full-head (*n* = 775), cleft palate patients (*n* = 4), patients with surgical plates or who had orthognathic surgery (*n* = 6), patients with orthodontic brackets (*n* = 15), implants (*n* = 6), nasion, condyle, or symphysis cut (*n* = 21), and patients with bite-blocks

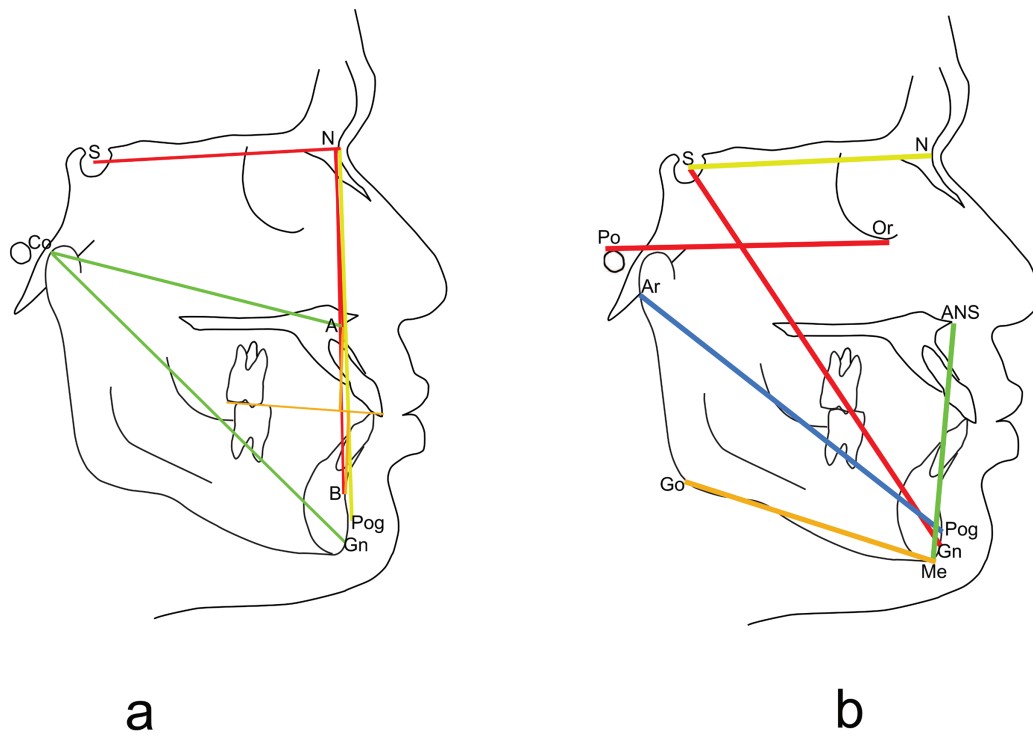

**Figure 2 Cephalometric landmarks.** Cephalometric landmarks (points and lines) used in the study (A) sagittal landmarks, (B) vertical landmarks.

**Table 1 Descriptive data analysis.**

| | Male | | | Female | | | P-value | Total | | |
|---|---|---|---|---|---|---|---|---|---|---|
| | Mean (SD) | Median | IQR | Mean (SD) | Median | IQR | | Mean (SD) | Median | IQR |
| Age (years) | 27.3 (11.7) | 25.0 | 11.7 | 37.9 (14.2) | 35.0 | 28.0 | 0.016* | 33.3 (14.0) | 28.0 | 17.5 |
| **CBCT volumetric measurements** | | | | | | | | | | |
| Right condyle (mm$^3$) | 926.1 (430.9) | 943.9 | 413.3 | 834.3 (310.4) | 815.0 | 443.0 | 0.443 | 874.0 (364.7) | 860.0 | 439.0 |
| Left condyle (mm$^3$) | 902.1 (464.5) | 830.0 | 381.5 | 884.4 (292.3) | 842.0 | 378.5 | 0.963 | 892.0 (370.7) | 835.0 | 361.0 |
| Both condyle (mm$^3$) | 1,828.1 (865.0) | 1,802.0 | 866.3 | 1,718.7 (521.5) | 1,628.0 | 471.5 | 0.69 | 1,766.0 (682.1) | 1,713.0 | 598.5 |
| Mandible volume (mm$^3$) | 41,713.1 (7942.4) | 43,424.5 | 12,341.3 | 43,832.4 (8,706.3) | 44,191.0 | 11,642.5 | 0.713 | 42,915.9 (8,338.3) | 43,425.0 | 11,941.5 |
| **Cephalometric sagittal measurements** | | | | | | | | | | |
| SNB (degrees) | 78.5 (5.9) | 79.2 | 7.15 | 78.8 (5.1) | 78.2 | 5.8 | 0.927 | 78.7 (5.4) | 78.7 | 6.15 |
| ANB (degrees) | 3.7 (4.1) | 4.2 | 3.43 | 6.3 (2.4) | 6.4 | 2.05 | 0.027* | 5.1 (3.5) | 5.9 | 3.7 |
| Wits appraisal (mm) | 0.36 (7.8) | 2.4 | 4.8 | 3.4 (3.4) | 3.4 | 4.7 | 0.425 | 2.1 (5.9) | 3.0 | 4.6 |
| Nasion perpendicular to Pog (mm) | 2.3 (10.0) | 1.5 | 13.6 | 0.92 (9.9) | −1.2 | 11.7 | 0.624 | 1.5 (9.9) | 1.0 | 11.95 |
| Effective mandibular length (CoGn) (mm) | 114.4 (10.2) | 113.2 | 8.55 | 108.0 (5.6) | 108.5 | 8.0 | 0.018* | 110.8 (8.4) | 110.4 | 7.95 |
| Maxillomandibular differential (CoGn minus CoA) (mm) | 28.6 (10.6) | 27.1 | 10.28 | 22.7 (3.3) | 22.7 | 5.3 | 0.066 | 25.3 (7.9) | 23.5 | 7.45 |

(Continued)

| | Male | | | Female | | | P-value | Total | | |
|---|---|---|---|---|---|---|---|---|---|---|
| | Mean (SD) | Median | IQR | Mean (SD) | Median | IQR | | Mean (SD) | Median | IQR |
| **Cephalometric vertical measurements** | | | | | | | | | | |
| Y axis (SGn-FH) (degrees) | 57.4 (4.4) | 56.4 | 6.33 | 57.7 (5.6) | 58.4 | 6.7 | 0.988 | 57.6 (5.1) | 57.6 | 6.65 |
| Mandibular plane angle 1 (PoOr-GoMe) (degrees) | 26.9 (6.9) | 27.1 | 7.18 | 25.6 (8.9) | 26.3 | 7.0 | 0.297 | 26.1 (8.0) | 26.4 | 8.4 |
| Mandibular plan angle 2 (NS-GoGn) (degrees) | 35.8 (7.1) | 34.7 | 10.9 | 33.9 (7.8) | 33.7 | 7.3 | 0.408 | 34.8 (7.5) | 34.0 | 9.6 |
| Mandibular plane angle 3 (PoOr-GoGn) (degrees) | 23.8 (7.3) | 23.5 | 6.6 | 22.8 (8.6) | 23.2 | 7.2 | 0.391 | 23.2 (7.9) | 23.2 | 7.5 |
| MP-SN (MeGo-SN) (degrees) | 38.9 (7.0) | 38.3 | 10.95 | 36.8 (8.1) | 36.3 | 7.25 | 0.399 | 37.7 (7.6) | 37.1 | 10.25 |
| Lower anterior facial height (ANS-Me) (mm) | 71.5 (8.3) | 70.4 | 7.45 | 66.1 (5.2) | 67.0 | 6.8 | 0.022* | 68.4 (7.2) | 67.7 | 8.8 |
| Gonial angle 1 (Ar, Pog, Jarabak Go) (degrees) | 122.3 (10.8) | 116.9 | 15.73 | 117.9 (9.4) | 117.0 | 8.9 | 0.358 | 119.8 (10.1) | 117.0 | 9.9 |
| Gonial angle 2 (tangent to posterior and inferior mandibular borders) (degrees) | 127.9 (9.6) | 125.3 | 16.2 | 123.4 (11.1) | 122.7 | 8.7 | 0.187 | 125.3 (10.6) | 123.4 | 12.4 |
| Gonial angle 3 (Me, Ar, constructed Go) (degrees) | 130.9 (8.2) | 127.1 | 14.1 | 126.4 (9.0) | 126.2 | 8.3 | 0.137 | 128.3 (8.8) | 126.4 | 8.6 |

**Notes:**
Descriptive data showing means, standard deviations (SD), medians, and interquartile ranges (IQR) for the measured variables of each gender and total samples.
* Statistically significant at 0.05 level of significance.

or a rapid palatal expander ($n = 2$). Ultimately, the final sample included 37 patients: 16 males (43.2%) and 21 females (56.8%). The mean age of patients was $27.3 \pm 11.7$ years and $37.9 \pm 14.2$ years, for males and females, respectively. There was a significant age difference between the genders, $p = 0.01$. There was no significant difference found between the genders for the right condylar ($p = 0.44$), left condylar ($p = 0.96$), both condyles ($p = 0.69$), and the mandible volumes ($p = 0.71$). Therefore, using the total sample including both genders was appropriate for volumetric measurements. All the measured parameters from cephalometric measurements and CBCT were analysed for descriptive statistics in which the mean, standard deviation (SD), and the median and interquartile range were calculated (Table 1). The value of ANB was significantly higher in females compared to males ($p = 0.027$). However, the average of effective mandible length and lower anterior facial height were significantly higher in males compared to females ($p = 0.018$, $p = 0.022$, respectively). The dependent variables for the study were the right condyle, left condyle, both condyles, and mandibular volumes.

Right condyle correlation was calculated with the independent variables for males, females, and for both genders. No significant correlation was found for the right condyle in males. Females had negative significant moderate correlation with maxillomandibular differential ($r = -0.483$, $p$-value = 0.027). When both genders were combined and correlation was computed between right condyle and independent variables, it was found that the maxillomandibular differential had negative, significant, but low correlation with the right condyle ($r = -0.354$ and $p$-value = 0.031).

**Table 2  Total condylar volume correlations.**

| | Male | | Female | | Total | |
|---|---|---|---|---|---|---|
| | r | *p*-value | r | *p*-value | r | *p*-value |
| **Cephalometric sagittal measurements** | | | | | | |
| SNB | −0.091 | 0.737 | 0.014 | 0.953 | −0.078 | 0.645 |
| ANB | 0.462 | 0.072 | −0.090 | 0.697 | 0.140 | 0.408 |
| Wits appraisal | 0.471 | 0.066 | −0.117 | 0.614 | 0.164 | 0.333 |
| Nasion perpendicular to Pog | −0.097 | 0.721 | 0.094 | 0.685 | 0.025 | 0.884 |
| Effective mandibular length (CoGn) | −0.015 | 0.957 | 0.112 | 0.630 | 0.045 | 0.790 |
| Maxillomandibular differential (CoGn minus CoA) | −0.534 | 0.033* | −0.504 | 0.020* | −0.431 | 0.008* |
| **Cephalometric vertical measurements** | | | | | | |
| Y axis (SGn-FH) | −0.153 | 0.572 | −0.234 | 0.306 | −0.184 | 0.276 |
| Mandibular plane angle 1 (PoOr-GoMe) | −0.172 | 0.524 | −0.344 | 0.126 | −0.202 | 0.231 |
| Mandibular plan angle 2 (NS-GoGn) | −0.165 | 0.542 | −0.229 | 0.319 | −0.108 | 0.526 |
| Mandibular plane angle 3 (PoOr-GoGn) | −0.165 | 0.542 | −0.229 | 0.319 | −0.255 | 0.127 |
| MP-SN (MeGo-SN) | −0.041 | 0.880 | −0.221 | 0.360 | −0.072 | 0.673 |
| Anterior facial height (ANS-Me) | −0.388 | 0.137 | −0.261 | 0.254 | −0.234 | 0.163 |
| Gonial angle 1 (Ar, Pog, Jarabak Go) | −0.335 | 0.204 | −0.355 | 0.114 | −0.313 | 0.060 |
| Gonial angle 2 (tangent to posterior and inferior mandibular borders) | −0.175 | 0.517 | −0.349 | 0.121 | −0.224 | 0.184 |
| Gonial angle 3 (Me, Ar, Constructed Go) | −0.306 | 0.249 | −0.379 | 0.090 | −0.274 | 0.101 |

**Notes:**
Correlation between CBCT total condylar volume with cephalometric measurements among males, females, and total samples.
* Statistically significant at 0.05 level of significance.

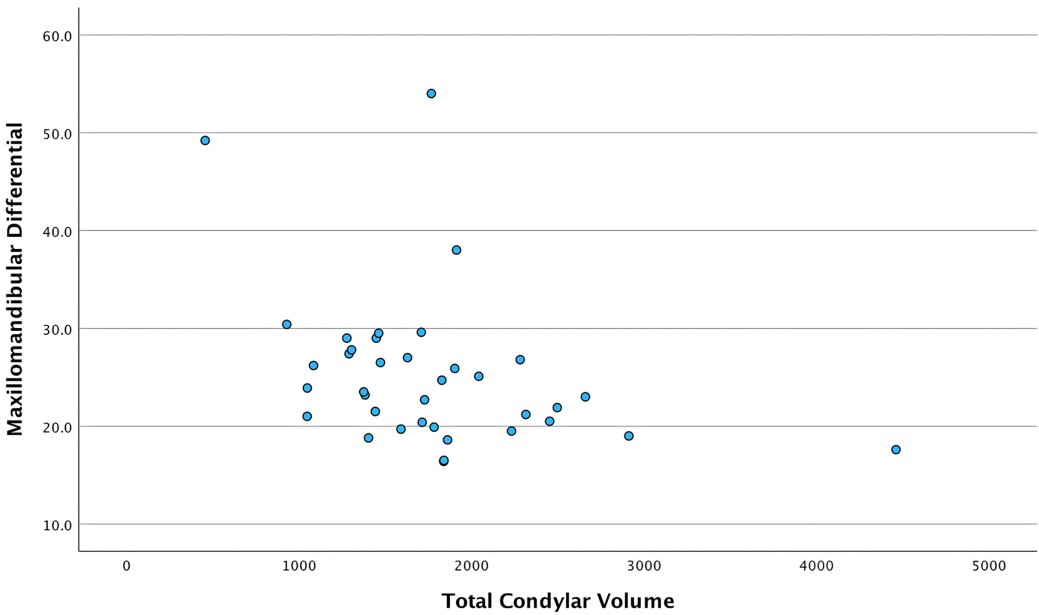

**Figure 3 Scatter plot showing the relationship between total condylar volume and maxillomandibular differential in the combined total sample.** Each point demonstrates the maxillomandibular differential value (Y-axis) corresponding to the total condylar volume (X-axis) in the combined sample.                                               

**Table 3 Mandibular volume correlations.**

| | Male | | Female | | Total | |
|---|---|---|---|---|---|---|
| | r | *p*-value | r | *p*-value | r | *p*-value |
| **Cephalometric sagittal measurements** | | | | | | |
| SNB | −0.215 | 0.425 | −0.043 | 0.854 | −0.139 | 0.411 |
| ANB | 0.435 | 0.092 | −0.034 | 0.882 | 0.166 | 0.327 |
| Wits appraisal | 0.521 | 0.039* | −0.021 | 0.929 | 0.225 | 0.180 |
| Nasion perpendicular to Pog | −0.279 | 0.295 | −0.162 | 0.484 | −0.227 | 0.176 |
| Effective mandibular length (CoGn) | 0.018 | 0.948 | 0.401 | 0.071 | 0.150 | 0.376 |
| Maxillomandibular differential (CoGn minus CoA) | −0.468 | 0.068 | −0.110 | 0.634 | −0.287 | 0.084 |
| **Cephalometric vertical measurements** | | | | | | |
| Y axis (SGn-FH) | 0.179 | 0.506 | 0.053 | 0.819 | 0.112 | 0.508 |
| Mandibular plane angle 1 (PoOr-GoMe) | 0.031 | 0.910 | −0.231 | 0.315 | −0.128 | 0.450 |
| Mandibular plan angle 2 (NS-GoGn) | −0.065 | 0.812 | −0.282 | 0.216 | −0.213 | 0.206 |
| Mandibular plane angle 3 (PoOr-GoGn) | −0.32 | 0.905 | −0.366 | 0.103 | −0.202 | 0.231 |
| MP-SN (MeGo-SN) | 0.044 | 0.871 | −0.249 | 0.277 | −0.174 | 0.304 |
| Anterior facial height (ANS-Me) | −0.226 | 0.399 | 0.109 | 0.640 | −0.072 | 0.673 |
| Gonial angle 1 (Ar, Pog, Jarabak Go) | −0.288 | 0.279 | −0.472 | 0.031* | −0.415 | 0.011* |
| Gonial angle 2 (tangent to posterior and inferior mandibular borders) | −0.160 | 0.553 | −0.474 | 0.03* | −0.334 | 0.043* |
| Gonial angle 3 (Me, Ar, Constructed Go) | −0.238 | 0.374 | −0.377 | 0.092 | −0.318 | 0.055 |

Notes:
Correlation between CBCT mandibular volume with cephalometric measurements among males, females, and total samples.
* Statistically significant at 0.05 level of significance.

When the left condyle correlation was investigated with the independent variables in males, there was significant moderate-to-strong correlation with ANB (r = 0.553, *p* = 0.026) and maxillomandibular differential (r = −0.596, *p* = 0.015). Meanwhile, no significant correlations were found in females. When both genders were combined, significant moderate correlation was found between the left condyle and maxillomandibular differential (r = −0.417, *p* = 0.01). Correlation between both condyles with the independent variables is summarized in Table 2. In males, females, and in both genders together, effective maxillomandibular differential had negative, moderate, and significant correlation with both condyles. Figure 3 shows the relationship between total condylar volume and maxillomandibular differential in the combined sample. There were no correlations for right condyle, left condyle, and both condyles' volumes with the vertical measurements.

Table 3 presents the correlation between the mandible volume and the independent variables. Regarding sagittal measurements in males, Wits had positive, moderate, and significant correlation with the mandible volume. However, no significant correlations were found between the mandibular volume and vertical measurements in males. In females and both genders combined, gonial angle 1 had an inverse, moderate, and significant correlation with the mandible volume. The same relationship existed in gonial

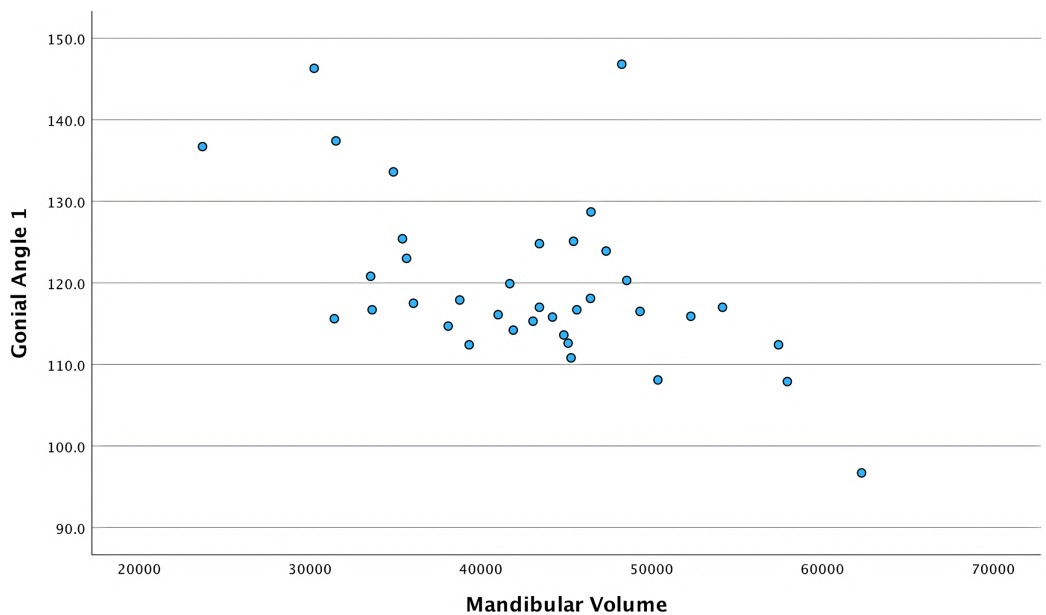

**Figure 4 Scatter plot showing the relationship between mandibular volume and gonial angle 1 in the combined total sample.** Each point demonstrates the gonial angle 1 value (Y-axis) corresponding to the mandibular volume (X-axis) in the combined sample.

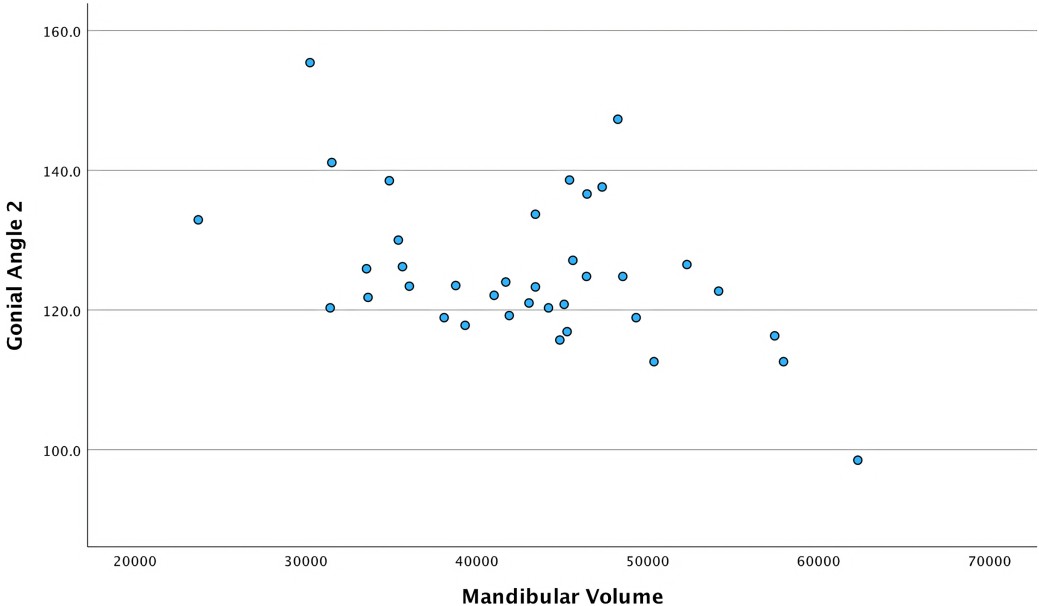

**Figure 5 Scatter plot showing the relationship between mandibular volume and gonial angle 2 in the combined total sample.** Each point demonstrates the gonial angle 2 value (Y-axis) corresponding to the mandibular volume (X-axis) in the combined sample.

angle 2, but it was moderate in females and weak in the combined genders sample. Figures 4 and 5 show the relationship between mandibular volume and gonial angles 1 and 2, respectively, in the combined sample.

## DISCUSSION

Assessment of the mandibular condyle and mandible is essential in craniofacial and orthodontic diagnosis. The mandible can have several growth patterns leading to different skeletal classifications in the sagittal and vertical dimensions. In the sagittal dimension, patients can have skeletal class I, II or III. They can be skeletally classified to have normal, retrognathic, or prognathic mandibles. Patients can also have vertically normal, increased, or decreased mandibular plane and gonial angles. The sagittal and vertical relationships between the condyles, mandible, and skeletal classes should be further investigated. Traditionally, this investigation was only done using 2D cephalometric measurements. These measurements had limited diagnostic capability depending on the plane that had been used. With the advent of CBCT technology, the evaluation of the mandibular measurements has evolved and 3D volumetric measurements can be employed.

To assign patients to each skeletal class, different landmarks on the craniomandibular complex are used. In this study, different analyses for classification including Downs, Steiner, Wits appraisal, and McNamara were used. The measurements taken from each analysis were compared to the following volumetric measurements: right and left condyle volumes, combined condylar volume, and mandibular volume. This comparison was used to assess the extent to which 3D volumetric measurements can be related to 2D cephalometric measurements.

The aim of this study was to find how 3D volumetric measurements of the condyles and mandible are correlated to the different skeletal classes and cephalometric measurements in both sagittal and vertical dimensions. Using cephalometric measurements, the ANB increased in females. This increase reflects the skeletal Class II discrepancy in the Saudi population (Al-Jasser, 2005), which was previously found to be associated with females compared to the skeletal Class III discrepancy in Jeddah City in Saudi Arabia (Alogaibi et al., 2020). Effective mandibular length and lower anterior facial height were found to be higher in males. This sexual dimorphism has also been reported in the literature (Duthie et al., 2007). This suggests that males have larger and therefore more prognathic mandibles.

Upon investigation, the relationship between the condylar volumes (per side) and 2D measurements (obtained using CBCT) were as follows: there was a moderate, negative correlation between the right condyle volume and the maxillomandibular differential in females; and a moderate, negative correlation between the left condyle volume and the maxillomandibular differential in males. It can be noted that the right condyle volume has significant correlation with maxillomandibular differential in females. Meanwhile, this relationship is significant on the left side in males. This difference in the significance of correlations between the right and left sides in the condylar volume may be related to the sexual dimorphism in the volume of the condyle as reported previously (Lentzen et al., 2022; Shetty et al., 2022) and to the inherent asymmetry in the dentofacial complex (Fischer, 1954). The relationships between the right condyle volume and the left condyle volume with the maxillomandibular differential in the combined-genders sample exhibited

significant negative correlations (weak correlation on the right side and moderate on the left side).

The total condylar volume had a moderate negative significant correlation with the maxillomandibular differential in males, females, and the combined samples. This finding highlights another relationship between the condyle and skeletal patterns in the sagittal dimension in addition to relationships found in previous studies. For example, in a study that evaluated the dimensions and shape of the condyle within the skeletal classes using the ANB value (*Mohsen et al., 2023*) it was found that condylar height was greater in class III than class I or class II skeletal patterns. This current study found that ANB is not significantly correlated to condylar volume, contrary to the study by *Mohsen et al. (2023)*. This could be due to the fact that *Mohsen et al. (2023)* evaluated only a two-dimensional variable (height). Meanwhile, the current study evaluated the total condylar volume. The previous study showed that the height and width of the condyles in males were greater than in females, which is similar to the findings in our study with respect to condylar volume. However, the differences between genders in the current study were not statistically significant.

Regarding the relationship between the combined volume of the condyles and vertical measurements, a prior study found that significant differences exist between hypodivergent and hyperdivergent patients in the anteroposterior and mediolateral condylar widths, along with significant differences in the condylar head angle and shape (*Park, Kim & Park, 2015*). The current study did not find significant correlations between the vertical measurements and the condylar volume. This difference in results can be attributed to the fact that the previous study also measured two-dimensional features, contrary to the 3D measurements used in the current study. Another study found that condylar volume tends to be larger in low mandibular plane angle patients. This is consistent with the current study that found a negative correlation between the total condylar volume and mandibular plane angles 1, 2, 3 and the MP-SN angle (*Saccucci et al., 2012*). It should be noted that this correlation did not reach statistical significance ($p > 0.05$).

Regarding the volume of the mandible in males, the sagittal measurement, namely Wits appraisal, showed a positive correlation with moderate strength. This finding shows that the mandible volume is related to the anteroposterior discrepancy within the maxillomandibular complex, but is not related to the cranial base (ANB value). With respect to this direct relationship with Wits appraisal, these findings could be improved by controlling the rotation of the occlusal plane. Therefore, it can be suggested that CBCT volumetric measurement of the mandible may reflect the sagittal skeletal class while accounting for confounding factors such as occlusal plane position. In females, with respect to Wits appraisal, a weak inverse relationship is seen in the mandible volume. This relationship occurs due to the fact that when the Wits appraisal value tends to be positive, it shows a smaller and more retrognathic mandible in the anteroposterior direction, which is a characteristic feature in females.

The volume of the mandible in females and in the total sample has significant negative correlation to the vertical measurements, namely the gonial angles 1 and 2. This finding

could be due to the fact that as the mandible volume increases, such as during aging, the gonial angles become more obtuse (*Xie & Ainamo, 2004*). It should be noted that the correlation is stronger in the gonial angle rather than in the mandibular plane angle in the vertical dimension. This may indicate that the aforementioned relationship is related to the maxillomandibular complex rather than being related to the cranial base.

From these results, it can be inferred that there is a stronger association between the total condylar volume and the maxillomandibular differential in the sagittal dimension. The mandibular volume has a greater association with the gonial angles 1 and 2, representing the vertical dimension. The volumetric measurements of the condyle and mandible are more dimensionally-related to skeletal classes, as evidenced by the maxillomandibular differential value, rather than being positionally-related, as evidenced when using the ANB value. The no-association null hypothesis of this study can be rejected because of these correlated variables in the sagittal and vertical dimensions. This study shows that the volumetric dimension should be considered in the evaluation of the skeletal classes of patients to provide better insight for orthodontic treatment planning and assessment of craniofacial structures.

One of the strengths of this study is that it uses both CBCT technology and cephalometry to achieve accurate results. The study evaluated measurements from multiple cephalometric analyses. Some of these measurements, such as mandibular plane and gonial angle, were measured using different landmarks to delineate small differences between these measurements. Unlike another study (*Shetty et al., 2022*), this study showed that using the 3D segmentation techniques, especially after removing structures other than the mandible, yielded better visualization of the results and correlations between volumetric and cephalometric techniques. Specifically, this current study segmented the condyle using a simpler technique to draw the line that passes the inferior most part of the sigmoid notch. The study relied on using the line perpendicular to the posterior border of the mandible itself instead of using the Frankfort plane (*Mohsen et al., 2023*; *Santander et al., 2020*; *Shetty et al., 2022*). This removes confounding factors, such as the position of the orbitale and the porion that are outside the mandible. Using this technique, the segmentation of the condyle is independent from structures other than the mandible and allows for condylar segmentation even when using CBCTs that do not show a full head view. Finally, the concept of "volumetric dimension" became more pronounced when analyzing these correlations with the sagittal and vertical dimensions.

Limitations of this study include that it is a single-center study with a relatively small sample size. Future studies could include patients from different countries. Conducting the same study in a multi-center setting could provide additional results showing the variation among different patient populations. The sample size could be increased to confirm the results on a larger scale; although this study had a total of 37 cases, which is larger than the number suggested by the sample size calculation ($n = 33$).

## CONCLUSIONS

In this study, total condylar volume correlated with maxillomandibular differential representing the sagittal skeletal classification. Mandibular volume was found to have

significant correlation with the gonial angles 1 and 2, representing vertical skeletal classification in females and in the total sample. It is suggested that the correlation between the total condylar volume and the maxillomandibular differential shows a more dimensional relationship, rather than a positional relationship to the maxillomandibular complex. Therefore, assessing the skeletal classes should include multiple dimensional measurements from all directions accompanied by volumetric analysis. This finding illustrates the importance of combining volumetric and cephalometric assessment in the field of orthodontics.

### Funding

This project was funded by the Deanship of Scientific Research (DSR) at King Abdulaziz University, Jeddah, under grant no. J: 207-165-1440. The funders had no role in study design, data collection and analysis, decision to publish, or preparation of the manuscript.

### Grant Disclosures

The following grant information was disclosed by the authors:
Deanship of Scientific Research (DSR): 207-165-1440.

### Competing Interests

The authors declare that they have no competing interests.

### Author Contributions

- Hussain Y. A. Marghalani conceived and designed the experiments, performed the experiments, analyzed the data, prepared figures and/or tables, authored or reviewed drafts of the article, and approved the final draft.

### Human Ethics

The following information was supplied relating to ethical approvals (*i.e.*, approving body and any reference numbers):

Research Ethical Committee (REC) at Faculty of Dentistry in King Abdulaziz University.

### Data Availability

The raw data are in the Supplemental File.

### Supplemental Information

Supplemental information for this article can be found online at http://dx.doi.org/10.7717/peerj.16750#supplemental-information.

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
