# Peer review of "Volumetric comparison of mandibular condyles and mandibles in the different skeletal classes in the Saudi population"

_PeerJ, doi:10.7717/peerj.16750_

## Round 0.1 · original submission · Minor Revisions

One of the reviewers also recommends the authors seeking language assistance of a professional academic language editing service.

**Language Note:** The Academic Editor has identified that the English language must be improved. PeerJ can provide language editing services - please contact us at copyediting@peerj.com for pricing (be sure to provide your manuscript number and title). Alternatively, you should make your own arrangements to improve the language quality and provide details in your response letter. – PeerJ Staff

Reviewer 1 ·

Basic reporting

No comment

Experimental design

No comments

Validity of the findings

No comments

Additional comments

Abstract: please reframe this sentence “A sample from middle eastern Saudi population consisting of 37 full-head CBCTs (74 condyles) for adult patients was examined”

Please reframe into simpler sentence- “Also, the volumetric dimension is correlated with the sagittal and vertical dimensions of the different orthodontic skeletal classes providing more information in addition to cephalometric measurements”

The introduction is written appropriately.

Materials and methods: Inclusion criteria-The authors can be a little bit clearer with middle eastern ethnicity and Saudi nationality (it seems to be overlapping and confusing)
Rest of the materials and methods is clear.

Results: Please check the format of figures 4 and 5 (compliant with journal formats)

Discussion: the authors can add a few points on the different methods of volumetric assessment of the condyles used in the other studies and compare them to the present study.

Limitations and conclusions are appropriate

References: most of the references are contemporary and follow the journal style of referencing.

Reviewer 2 ·

Basic reporting

The author should reviewed English in order to make clearer the article. The structure of the article follows the standards of this type of paper, providing appropriate references. Moreover, tables and figures can be of help to better understand the research.
In spite of these, the author should express if maxillo mandibular malformations as condylar hyperplasia are included. In addition, it could be interesting, if the author will add figures showing 3D measurement and 3D dolphin’s evaluation.

Experimental design

This reasearch is not original. Similar studies were already published, but no one had focused attention on Saudi population before.

The model of study is replicable.

Validity of the findings

no comment

Reviewer 3 ·

Basic reporting

The manuscript is well-structured and easy to read through. However, there are several mistake in the English languahe, i.e. "To determine if condylar volume and mandibular volume are related to the skeletal classes...". "If" should be replaced with the term "whether". I suggest the authors to optimize the manuscript with some kind of language editing services. Furthermore, I would shorten the Introduction to make it more concise, with less literature review but rather a literature overview.

Experimental design

The methoodlogy of the study is in my view able to address the research question that the author posed. Also adequate statistics have been used.

Validity of the findings

In Table 1 I see that the condylar volume of female tended to be smaller compared to male whereas the mandibular volume of female was larger than that of male. Can you elaborate on this? From previous literature, there is a positive correlation between condylar volume and mandibular volume. Also the distance from nasion to pogonion is smaller in female compred to male. Does this finding suggest that female have more protruded chin? Can the author further explain the left right differences that were present?

Additional comments

Does figure 1 represents a well-segmented mandible? I notice that the roots and even cervical portions of crowns to be present in the lower incisal region. Did the author really measure the mandibular volume of was also a part of the volume of the dentition being measured? Can the author comment on this?

---

## Round 0.2 · accepted · Accept

Thank you for making the revision of your original manuscript and for addressing all of the reviewers' comments. In my opinion the manuscript is now ready for publication.